# Potency and Selectivity of SMAC/DIABLO Mimetics in Solid Tumor Therapy

**DOI:** 10.3390/cells9041012

**Published:** 2020-04-18

**Authors:** Xiao-Yun Zhao, Xiu-Yun Wang, Qi-Yao Wei, Yan-Ming Xu, Andy T. Y. Lau

**Affiliations:** Laboratory of Cancer Biology and Epigenetics, Department of Cell Biology and Genetics, Shantou University Medical College, Shantou 515041, China; 18xyzhao@stu.edu.cn (X.-Y.Z.); 19xywang@stu.edu.cn (X.-Y.W.); 19qywei@stu.edu.cn (Q.-Y.W.)

**Keywords:** inhibitor of apoptosis protein (IAP), second mitochondria-derived activator of caspase (SMAC), direct IAP-binding protein with low pI (DIABLO), SMAC mimetics (SMs), clinical trial, therapy, solid tumor

## Abstract

Aiming to promote cancer cell apoptosis is a mainstream strategy of cancer therapy. The second mitochondria-derived activator of caspase (SMAC)/direct inhibitor of apoptosis protein (IAP)-binding protein with low pI (DIABLO) protein is an essential and endogenous antagonist of inhibitor of apoptosis proteins (IAPs). SMAC mimetics (SMs) are a series of synthetically chemical compounds. Via database analysis and literature searching, we summarize the potential mechanisms of endogenous SMAC inefficiency, degradation, mutation, releasing blockage, and depression. We review the development of SMs, as well as preclinical and clinical outcomes of SMs in solid tumor treatment, and we analyze their strengths, weaknesses, opportunities, and threats from our point of view. We also highlight several questions in need of further investigation.

## 1. Introduction

In cancer biology, resistance to cell death is one of the hallmarks of cancer. Programmed cell death (PCD), especially apoptosis, functions as a natural defense to cancer development [1]. Afterward, targeted drugs to promote PCD emerged, exemplified by apoptosis-inducing compounds, showing their potential in cancer therapy [2]. The inhibitor of apoptosis proteins (IAPs) are a class of apoptosis regulators, which present a crucial role in the control of cell survival by suppressing both initiator and effector caspases [3,4]. Second mitochondria-derived activator of caspase (SMAC) could promote caspase activation by binding to IAPs and removing the inhibition of IAPs to caspases [5]. Direct targeting of IAPs by using mimetics of its natural antagonist SMAC can synergize the effect of traditional cancer therapy in myeloid malignancies, e.g., acute myeloid leukemia (AML) and chronic myeloid leukemia (CML) [6]. In this review, we focus on the application of SMAC mimetics (SMs) in solid tumor therapy. By analyzing the potential mechanism of endogenous SMAC inefficiency, summarizing the latest preclinical and clinical outcomes of SMs in solid tumor treatment, and updating the cell death pathways regulated by SMs, we wish to deliver insight into the further application of SMs in cancer therapy.

## 2. SMAC in Cancers

### 2.1. Reduced SMAC Expression/Promoted SMAC Degradation

Perturbation of apoptosis is a vital cause in carcinogenesis and tumor progression. In the process of apoptosis, SMAC plays an important part. SMAC was identified as a detergent-soluble factor, which can stimulate caspase-3 activation in the presence of the known water-soluble factors Apaf-1, cytochrome c, and procaspase-9 in 2000 [5]. After that, it was found to be identical to direct IAP-binding protein with low pI (DIABLO) according to their sequences in 2000 [7]. SMAC/DIABLO is ubiquitously expressed in normal tissues, and the expression of SMAC messenger RNA (mRNA) in adult testis is the highest [8].

According to previous research data, the expression of SMAC is altered in many malignant tumors compared with normal tissues. The expression of SMAC is downregulated in renal cell carcinoma [9,10], colorectal cancer [11], bladder cancer [12,13], lung cancer [14], hepatocellular carcinoma [15], and testicular germ cell tumors [16]. Heat-shock transcription factor 1 (HSF1) downregulates SMAC expression in pancreatic cancer cells, thus promoting pancreatic cancer [17]. On the other hand, SMAC is also overexpressed in some tumors. High SMAC expression is linked to early local recurrence of cervical cancer [18], and additional non-apoptotic functions of SMAC are related to regulating phospholipid synthesis essential for cancer growth and development [19]. In addition, SMAC can be upregulated by transcription factor E2F1 (E2F1) in lung cancer at both mRNA and protein levels [20]. As stated in the chart calculating the datasets from The Cancer Genome Atlas (TCGA) program, downloaded from UALCAN (a comprehensive, user-friendly, and interactive web resource for analyzing cancer omics data) [21], SMAC is generally upregulated in cancers at the mRNA level (Figure 1a).

For one thing, SMAC is a pro-apoptotic protein that counteracts the inhibitory activity of IAPs leading to activation of caspases and apoptosis [7,22,23]. SMAC was described as an inhibitor of XIAP in the beginning, binding to the baculoviral IAP repeat (BIR) domains, BIR 2 and BIR 3 [24]. Then, some researchers found that SMAC can selectively induce the degradation of cellular inhibitor of apoptosis 1 and 2 (cIAP1 and cIAP2) in HeLa cells, but not XIAP [25]. In other researches, SMAC3, a SMAC splicing variant, was shown to have the ability to induce auto-ubiquitination and destruction of XIAP [8]. Like many RING domain-containing proteins, XIAP, cIAP1, and cIAP2 possess ubiquitin ligase activity toward themselves and other target proteins [26,27,28,29]. Likewise, cIAP1 (BIR-containing protein 2, BIRC2) [30], cIAP2 (BIRC3) [30], XIAP (BIRC4) [26,31,32], Livin (KIAP/ML-IAP/BIRC7) [33], and Bruce (Apollon/BIRC6) [34] were demonstrated as ubiquitin-protein ligases for SMAC. These results show that IAPs can promote SMAC degradation via the ubiquitin-proteasome pathway.

Although SMAC is more likely to be upregulated than downregulated at the mRNA level in cancers, its pro-apoptotic ability might have a weak role because of degradation. As a result of either lower expression or higher degradation, the low level of SMAC could cause an increased apoptotic threshold which would allow cancer cells to develop an enhanced resistance to novel clinical treatments aiming at inducing apoptosis. This is why a low level of SMAC is associated with shorter survival [10], resistance to therapies [12], or poorer prognosis [11,13,14].

### 2.2. Blockage of SMAC Release

SMAC could promote caspase activation by removing the inhibition of IAPs. Only mature SMAC has this activity, while its precursor with an intact signal sequence does not. At the N-terminus of SMAC, there is a stretch of amino acids characteristic of mitochondrial targeting sequences that are normally removed from SMAC upon import into mitochondria [5]. When released from mitochondria, SMAC acts as a targeted molecule.

Proteins of the apoptosis regulator BCL-2 (BCL-2) family control intrinsic/mitochondrial apoptotic pathway by regulating mitochondrial outer membrane permeabilization (MOMP) [35,36]. MOMP allows the release of mitochondrial proteins from the mitochondrial intermembrane space (IMS) into the cytosol, including cytochrome c, SMAC, and HtrA2/Omi [7,37,38,39]. Upon stimuli, effector proteins, such as apoptosis regulator BAX/BCL-2-like protein 4 (BAX), BCL-2 homologous antagonist/killer (BAK), and BCL-2-related ovarian killer protein (BOK), are activated and oligomerize at the mitochondria outer membrane to mediate MOMP [36,40,41]. The anti-apoptotic/pro-survival BCL-2 proteins, including apoptosis regulator BCL-2 (BCL2), BCL-2-like protein 1 (BCL2L1), BCL-2-like protein 2 (BCL2L2), induced myeloid leukemia cell differentiation protein MCL1 (MCL1), and BCL-2-related protein A1 (BCL2A1), suppress cell death by binding and inhibiting BAX and BAK [36]. The direct activator “BH3-only” proteins, BH3-interacting domain death agonist (BID) and BCL-2-like protein 11 (BIM), can directly induce BAK and BAX oligomerization and MOMP. The de-repressor “BH3-only” proteins, i.e., BCL-2-associated agonist of cell death (BAD), BCL-2-interacting killer (BIK), BCL-2-modifying factor (BMF), activator of apoptosis Harakiri (HRF), phorbol-12-myristate-13-acetate-induced protein 1 (Noxa), and BCL-2 binding component 3 (PUMA/BBC3), mainly interact with the anti-apoptotic proteins to promote apoptosis [42,43].

Although the BCL-2 family is involved in many diseases, the most distinguished one is cancer. The most potent and ubiquitous mechanism is overexpression of anti-apoptotic members [44]. From the chart downloaded from UALCAN [21] (Figure 1b–d), some anti-apoptotic BCL-2 proteins are visibly upregulated in some cancers. In this case, no matter whether the levels of BAX, BAK, and/or BOK are high or not, the MOMP-mediated release of SMAC would be relatively blocked.

Furthermore, there are other studies showing inhibitors of SMAC release. Survivin, another member of IAPs, can associate with SMAC in mitochondria to delay its release and stabilize the cytosolic levels of released SMAC in cytosol [45]. From the chart downloaded from UALCAN [21] (Figure 1e), baculoviral IAP repeat-containing protein 5 (survivin/BIRC5) is overexpressed in many cancers making it more possible to regulate the release of SMAC. This also supplements the interaction between SMAC and IAPs. In addition, heat-shock protein 27 (HSP27) can also inhibit the release of SMAC, thereby allowing increased survival and drug resistance in multiple myeloma (MM) cells [46].

### 2.3. Loss of IAP Binding Ability in Mutated SMAC

After SMAC is localized in the cytoplasm, its IAP-binding motif (IBM), which consists of four N-terminus amino-acid residues (Ala–Val–Pro–Ile), also known as the AVPI segment, interacts with BIR domains of IAPs [47,48,49]. SMAC dimerization is essential to its function in activating procaspase-3, promoting the enzymatic activity of mature caspase-3 [50], and relieving XIAP-mediated caspase inhibition [51]. It was proven that mutations of the first amino acid lead to loss of interaction with IAPs and concomitant loss of SMAC function [47,50], and that mutations of hydrophobic residues at the interface disrupt the dimer formation [50]. We can see that mutations of SMAC can occur in many types of cancers (Figure 2a). Although the alterations of SMAC/DIABLO occurred in 95 samples out of 10,950 patients with cancer (0.9%) (Figure 2b) [52,53], this still might be a potential mechanism of SMAC dysfunction. It was shown that c-Jun N-terminal kinase 3 (JNK3) can phosphorylate SMAC. JNK3-mediated phosphorylation of SMAC markedly attenuates the interaction between SMAC and XIAP [54]. Thus, both mutations and phosphorylation might affect SMAC by changing its protein structure.

Furthermore, one study reported that functional mutation of SMAC underlies human progressive hearing loss, designated as autosomal-dominant nonsyndromic hearing loss 64 (DFNA64), by causing degradation of mutant and wild-type SMAC and leading to mitochondrial dysfunction [55].

### 2.4. Depression of SMAC Activity Due to Overexpression of IAPs

The pro-apoptotic activity of SMAC mainly acts on XIAP, cIAP1, and cIAP2. There is evidence to show that the average level of XIAP expression is higher in renal cell carcinoma compared to an autologous normal kidney [56], even increasing beyond that of SMAC [57]. IAPs are overexpressed in various solid and hematological malignancies, and high expression levels of IAPs are associated with resistance to standard chemotherapeutics and radiation therapy, as well as poor prognosis [4,58,59]. SMAC activity is depressed in the case of increased XIAP, cIAP1, or cIAP2. It is worse when other IAPs co-overexpress, as they compete with XIAP, cIAP1, or cIAP2 for binding to SMAC and, in this way, preserve them from inhibition by SMAC [60,61,62]. From the chart downloaded from UALCAN [21] (Figure 1f–h), we can also observe the overexpression of XIAP, cIAP1, and cIAP2 at the mRNA level in some types of cancer. Likewise, as mentioned above, IAPs can promote SMAC degradation through the ubiquitin-proteasome pathway.

In brief, as a result of either low level or dysfunction, the activity of SMAC decreases in cancer cells, rendering it unable to antagonize the upregulated IAPs. Thus, SMs, also known as IAP antagonists, which are a new class of targeted drugs to suppress the IAPs, came into view [63].

## 3. Mechanism of Action of SMs

SMs induce cancer cell death predominantly through a cIAP-dependent mechanism regulated by death receptor ligands, such as tumor necrosis factor alpha (TNFα) [64] (Figure 3). Binding of SMs to cIAP1 and cIAP2 leads to a conformational change of these IAP proteins, inducing its auto-ubiquitination and subsequent proteasomal-mediated degradation [65,66,67]. Degradation of cIAP1 and cIAP2 in cancer cells allows for the accumulation of nuclear factor-κB (NF-κB)-inducing kinase (NIK) and stimulates the activation of the non-canonical NF-κB pathway [67]. This stimulation of the NF-κB pathway causes an autocrine synthesis of cytotoxic cytokines like TNFα, which subsequently engages the TNF receptors such as TNF-R1 [68]. With depletion of cIAPs, TNFα cannot stimulate receptor-interacting protein 1 (RIP1) ubiquitination; subsequently, RIP1 that is released from the TNF-R1 complex assembles the death-inducing complex IIa, containing RIP1, tumor necrosis factor receptor superfamily member 6 (FAS)-associated protein with death domain (FADD), and caspase-8, which triggers caspase-8 activation and the TNFα-induced apoptotic pathway. This partly explains the combination of SMs with either chemo- or radiotherapy, which enhances the expression of TNFα, revealing synergistic activity [69]. Moreover, XIAP can also influence TNFα signaling, activate NF-κB, and block apoptosis at the effector phase, which is a point where multiple signaling pathways converge. Moreover, XIAP is a potent suppressor of TNF-related apoptosis-inducing ligand (TRAIL)/FasL-induced cell death by interacting with caspase-3, -7, or -9 [70,71]. Hence, treatments targeting XIAP may especially help to overcome resistance to chemotherapeutic agents. SMs directly bind XIAP, thereby activating caspases and inducing apoptosis of cancer cells [72]. Taken together, releasing caspases from the inhibitory interaction with IAP proteins and inducing apoptosis through extrinsic and intrinsic pathways can also provide a possible explanation for the phenomenon that the treatment of SMs sensitizes cancer cells to conventional chemo- and radiotherapy [73]. In other words, SMs, as sensitizers, reduce the threshold for cell death induced by chemo- or radiotherapy, which directly provoke cell death pathways. In situations in which a functional caspase-8 is deficient, SMs were reported to trigger the production of necrosome, consisting of RIP1, receptor-interacting protein kinase 3 (RIP3), and mixed lineage kinase domain-like (MLKL), thereby promoting cancer cells to undergo TNFα-stimulated necroptosis [74]. In the absence of functional caspase-8, the RIP1-containing complex IIa cannot be formed, allowing non-ubiquitinated RIP1 to interact with RIP3 through their RIP homotypic interaction motif [75]. Subsequently, activated RIP3 binds to and phosphorylates MLKL forming the necrosome, allowing necrotic cell death to take place.

Even though the vast majority of studies suggested that SM-facilitated cell death needs autocrine TNFα production, it was reported that SM-induced cell death also occurred in a TNFα-independent manner. The ripoptosome that contains RIP1, FADD, caspase-8, and cellular FADD-like interleukin beta-converting enzyme (FLICE)-like inhibitory protein (c-FLIP) can promote either apoptosis or necrosis [76,77]. c-FLIP is mainly expressed as two splice forms in human cells: a long form c-FLIP_L_ and a short form c-FLIP_S_, and both of them exhibit their oncogenic function primarily by inhibiting caspase-8 activation [78]. Furthermore, the assembly of the ripoptosome is controlled, amongst others, by the two major isoforms of cFLIP. c-FLIP_L_ prevents ripoptosome formation, but c-FLIP_S_ promotes the assembly of ripoptosome [77]. When RIP1 level is low or cFLIP_L_ level is high, cells are resistant to the formation of the ripoptosome, leading to death. Intriguingly, in the loss of cIAPs, c-FLIP_S_ was able to protect cells from ripoptosome-induced apoptosis but evoked necroptosis, which is caused by the absence of caspase activity within the complex. In addition to their effect on the tumor cells, SM-induced loss of cIAP1/2 in immune cells can activate the alternative NF-κB pathway, which promotes B-cell survival and provides a broad co-stimulatory signal to dendritic cells and T cells [79]. These signals can further induce an immune-modulatory activity, resulting in the massive secretion of proinflammatory cytokines against cancer cells [80,81]. However, high-dose SM treatment may have side effects that can lead to systemic toxicity, to cytokine release syndrome, or to blunting of tumor responses to the death ligands.

In principle, SMs can kill cancer cells via multiple cell death pathways; thus, the anticancer effect of SMs is theoretically excellent. In fact, only a small subset of cancer cell lines tested to date appear to be effectively killed by single-agent SMs [82,83,84]. Lacking a mechanistic understanding of this resistance is a leading hurdle to the effective application of these drugs. Thus, a mass of work concentrated on the key mechanisms of therapeutic resistance to SMs in cancer cells. Previous studies confirmed that some cancer cells evade SM-induced cell death via stabilization of cIAP2 [85,86,87]. Some resistant cell lines have such an intense tendency to upregulate cIAP2, while sensitive cells seem to be less susceptible to signals that upregulate cIAP2. Accordingly, chemical inhibitors of phosphatidylinositol 3-kinase (PI3K) were confirmed to hamper cIAP2 upregulation and could, therefore, sensitize H1299 cells to cell death induced by SMs alone [83]. A recent study found that SM-induced degradation of cIAP2 relies on not only binding to tumor necrosis factor receptor-associated factor 2 (TRAF2), but also the presence of cIAP1 [66]. In the absence of cIAP1 or TRAF2, the newly generated cIAP2 is no longer downregulated by SMs. Another report showed that overexpression of ubiquitin-specific protease 11 (USP11) led to stabilization of cIAP2 and promoted SM therapeutic resistance [86].

In addition, research showed that SM-induced loss of cIAP1/2, two critical regulators of the TNF receptor superfamily and NF-κB signaling, sensitizes cancer cells to TNFα- or TRAIL-mediated death [72]. Therefore, we suppose that a high level of TRAIL or TNF in cancer might make SMs more efficient. From the chart downloaded from UALCAN (Figure 1i–j), we can see that TRAIL and TNF do have high mRNA levels in a few types of cancer. On the other hand, studies showed that the deficiency of TNFα caused poor therapeutic efficacy of SMs [88,89]. According to recent research, the cell surface protein leucine-rich repeats and immunoglobulin-like domains protein 1 (LRIG1) contributes resistance to SMs by upregulating receptor tyrosine kinase (RTK) signaling and attenuating TNFα expression [90]. Furthermore, the interferon regulatory factor 1 (IRF1) was investigated to be critical for inducing the production of SM-mediated TNFα [91]. Several recent studies showed that multiple cancer lines of different origins, including glioblastoma, melanoma, ovarian adenocarcinoma, pancreatic carcinoma, and breast cancer, were resistant to SM-induced death in the absence of the transcription factor specificity protein 3 (SP3), which promotes autocrine TNFα expression [92,93]. Based on the above, one can draw a conclusion that the presence of TNFα is pivotal in ensuring the excellent efficacy of SMs. Thus far, levels of TNFα would likely be a momentous desirable factor for pinpointing patients who will benefit from single-SM treatment.

However, high expression of TNFα is necessary but not sufficient for inducing apoptosis via this kind of treatment. In other words, many cancer cell lines are resistant to SM-mediated apoptosis with high expression of TNFα. Taking malignant pleural mesothelioma (MPM) as an example, even in MPM patients with high levels of TNFα, MPM cell lines assessed were exceedingly resistant to SMs either alone or when incubated in the presence of clinically relevant levels of TNFα [94]. Further studies revealed that SM sensitivity could be restored by downregulating c-FLIP, which is an enzymatically dead caspase-8 homolog preventing caspase-8-mediated apoptosis. Similar studies established that downregulating the anti-apoptotic protein c-FLIP is crucial for the susceptibility of breast cancer cell lines to SMs [95].

## 4. Development and Clinical Trials of SMs

### 4.1. Development of SMs

In general, SMs can be divided into monovalent and bivalent. Monovalent SMs are composed of one SMAC-mimicking unit, whereas bivalent or dimeric SMs comprise two units. Compared with the monovalent SMs, most of the existing bivalent SMs exhibit higher binding affinities, as well as an ability to bind to the BIR 2 and BIR 3 domains of XIAP, resulting in the simultaneous activation of caspase-3/7. Cong et al. focused on the research articles of the past 15 years and clearly summarized the structural interaction between IAPs and SMAC, four generations of SMs, and representative antagonists in clinical evaluations [96]. Subsequently, Zhu et al. showed detailed strategies for designing bivalent small-molecule SMs and the progress in using them to antagonize IAPs, as well as their clinical potential [97]. These latest summaries are enormously helpful to guide further study to optimize the properties of bivalent SMs to ensure good bioavailability and targeted accumulation in various types of tumors.

### 4.2. SMs in Therapies

In initial exploration, earlier studies showed that cell-permeable SMAC-based peptides as single agents could not induce apoptosis in tumor cells. However, they are capable of potentiating the anticancer activity of other agents [98,99,100]. In 2004, Oost and colleagues showed that SMAC peptide-mimetics inhibited cell growth in seven different cell lines with diverse tumor types by screening various cancer cell lines [101]. The bivalent SMs were initially shown to potentiate the activity of TRAIL and TNFα but had no activity as a single agent in the T98G glioma cell line [102]. In the subsequent studies, both monovalent and bivalent SMs were shown to be effective in inhibition of cell growth and induction of apoptosis in cancer cell lines [103,104,105,106]. These data with different SMs from a number of laboratories provided the evidence that SMs may have the potential for the treatment of human cancer, even when used as single agents.

#### 4.2.1. Combined with Death-Inducing Ligands

As crucial antagonists of IAPs, SMs may effectively potentiate the antitumor activity of other anticancer agents. Earlier studies in 2002 showed that short SMAC peptides could synergize with the TRAIL receptor to enhance apoptosis in various tumor cells and suppress tumor growth in vitro and in vivo [98,99,100]. TRAIL is a member of the TNFα family; however, unlike TNFα, TRAIL shows very low toxicity to normal cells and tissues and is well tolerated in clinical trials [107]. Similarly, TRAIL has minimal anticancer activity used as monotherapy in clinical trials, and this limited its clinical development. Hence, the combination of SMs with TRAIL seems to be a particularly attractive strategy. Indeed, the combination of SMs with TRAIL evinced synergistic activity in numerous investigations. It was found that the SMAC peptides strongly enhanced the antitumor activity of TRAIL in an intracranial malignant glioma xenograft model in vivo, and the combination achieved eradication of established tumors without detectable toxicity to normal brain tissue [98]. In 2004, a bivalent SM was highly effective in potentiating apoptosis when combined with TRAIL and TNFα but had no activity as a single agent in the T98G glioma cell line [102]. In later studies, SMs were also shown to enhance apoptosis induced by TRAIL in pancreatic carcinoma models in vitro and in vivo [108,109,110]. SMs were reported to decrease TRAIL-stimulated invasion and metastasis, even in cancers such as cholangiocarcinoma (CHOL) [111]. Moreover, it is highly synergistic with TRAIL in vitro in both TRAIL-sensitive and TRAIL-resistant cancer cell lines of breast, prostate, and colon cancer [112].

In addition to TRAIL, strong synergy was also observed between SMs and TNFα in various cancer cell lines, such as glioma, breast cancer, lung cancer, and so on [82,102,113]. A later study indicated that SMs can affect 48% of cell lines when combined with TNFα, which was found to be ineffective in induction of apoptosis of 51 different cancer cell lines [95]. In addition, birinapant, an SM, in combination with TNFα, was reported to inhibit the growth of melanoma, including the serine/threonine-protein kinase B-raf (BRAF) inhibitor-resistant cell line [114]. Earlier studies showed that, in apoptosis-resistant leukemia cells, SMs can potentiate TNF-induced necroptosis by enhancing the formation of the necrosome complex [84,115]. Subsequently, SM treatment induced massive cell death and led to regression of tumors in solid tumors [116,117]. These findings indicated that SMs can combine with death receptors to trigger apoptotic cell death.

#### 4.2.2. Combined with Kinase Inhibitors

SMs were reported to potentiate the effects of epidermal growth factor receptor (EGFR, also known as ERBB1). For example, SMs targeting multiple IAPs promote apoptosis in response to the ERBB antagonists, trastuzumab, lapatinib, or gefitinib, in Her2-overexpressing breast cancer cells, or gefitinib in EGFR-overexpressing breast cancer cells [118]. Another study provided evidence that SMs display antitumor and anti-metastasis effects in vivo, and they contribute to EGFR inhibition and the reduction of its downstream mediators [119]. Furthermore, in glioblastoma, the SM LBW242 can synergize with several receptor tyrosine kinase inhibitors, including imatinib, nilotinib, nnp-aww541, and PKI166, to promote apoptosis [120]. Similarly, the simultaneous administration of SMs with bortezomib potently triggers apoptosis in a melanoma cell line [121].

#### 4.2.3. Combined with Chemotherapy

SMs were shown to sensitize various cancers toward chemotherapy. Since the first preclinical evidence that SMs potentiate the effects of various chemotherapeutics, there are more and more studies on the synergistic anticancer effect of SMAC peptides and chemotherapeutic drugs. In 2002, there was a study demonstrating that the SMAC peptide can enhance the induction of apoptosis and long-term antiproliferative effects of diverse antineoplastic agents including paclitaxel, etoposide, 7-ethyl-10-hydroxycamptothecin (SN-38), and doxorubicin in breast cancer and immortalized cholangiocyte cell lines [99]. Further study demonstrated that a cell-permeable SMAC peptide selectively reversed the apoptotic resistance of H460 lung cancer cells and, in combination with taxol and cisplatin, regressed the tumor growth in vivo with little toxicity to the mice [100]. Subsequently, combination of the SM JP1201 with gemcitabine induced regression of tumors in orthotopic xenograft and syngeneic tumor models, and prolonged survival in xenograft and transgenic models of pancreatic cancer [122]. A subsequent study also demonstrated this for synergistic sensitization of non-small-cell lung cancers to standard chemotherapy agents [123].

Series of SMs were proven to be able to enhance the antitumor effects in different cancer cell lines when combined with chemotherapy drugs in vitro. In 2011, it was demonstrated that a small-molecule SM, at nanomolar concentrations, significantly sensitized HNSC cells to gemcitabine-induced apoptosis and restored gemcitabine sensitivity in SMAC knockdown cells [124]. Then, LBW242 was found to strongly synergize with platin and taxol anticancer drugs that are commonly used in clinic in inducing apoptosis of ovarian cancer cells [125]. In 2017, it was reported that a synergistic combination of SM BV6 and bortezomib effectively triggers cell death in B-cell non-Hodgkin lymphoma cells, even when apoptosis is blocked [126]. In addition, the combination of birinapant with norcantharidin (NCTD) was reported to promote anticancer activity in breast cancer cells [127]. Of note, some SMs were reported to be able to reverse resistance to chemotherapy agents by inducing cell death depending on the cell lines. It was demonstrated that treatment of SM LCL161 not only increases the effects of vincristine in neuroblastoma cells, but it is also able to overcome vincristine resistance in neuroblastoma cells [128]. Subsequently, SM Debio 1143 in vitro was demonstrated to inhibit the cell viability of two carboplatin-sensitive cell lines (IGROV-1 and A2780S), as well as three carboplatin-resistant cell lines (A2780R, SKOV-3, and EFO-21) [129].

Significantly, SMs in combination with chemotherapy in different kinds of tumor models were also explored. In 2013, there was a study reporting that SM-164 in combination with gemcitabine increases the number of apoptotic and dead pancreatic cancer cells, as well as inhibits tumor xenograft growth in nude mice [130]. Then, patient-derived xenograft models showed that the activity of a variety of chemotherapeutic drugs is potentiated by birinapant, and tumor growth in multiple primary patient-derived xenotransplant models is inhibited by birinapant at well-tolerated doses [131]. In addition, a subsequent study showed that Debio 1143-containing combinations effectively inhibit the growth of lung adenocarcinoma both in vitro and in vivo [132]. In particular, a study showed that the combination of LCL161 with chemotherapy regimens such as rituximab, gemcitabine, or vinorelbine exhibits synergistic antitumor activity in models of resistant lymphoma both in vitro and in vivo, but not in rituximab-sensitive cell lines in vivo [133]. Similarly, results of a recent study showed that, in the osteosarcoma model, co-treatment with LCL161 and doxorubicin is particularly effective, impeding primary tumor growth and delaying or preventing metastasis; however, it only efficiently kills osteosarcoma cells when TNFα is supplied in vitro [134].

#### 4.2.4. Combined with Radiotherapy

SMs were reported to enhance radiosensitivity in several types of cancer cell lines, such as colorectal cancer, glioblastoma, esophageal carcinoma (ESCA), and head and neck squamous cell carcinoma (HNSC). An early study showed that in vitro and in vivo radiosensitization of colorectal cancer HT-29 cells by JP1201 decreased the survival of HT-29 cells and tumor growth via an additive effect in apoptosis, as well as a reduction in the level of XIAP and an impairment of DNA repair mechanisms [135]. Subsequently, another study reported that a SM compound, SM-164, can enhance the radiosensitization of HNSC via activation of caspases [136]. Similarly, LCL161 acts as a strong radiosensitizer in human ESCA cells by inhibiting the expression of cIAP1 and promoting the activation of caspase-8, leading to enhanced apoptosis [137]. BV6 was shown to stimulate NF-κB activation, which was required for the SM-conferred radiosensitization of glioblastoma cells [138]. In colorectal cancer cells, BV6 also sensitizes cells to ionizing radiation by interfering with DNA repair processes and enhancing apoptosis [139]. Furthermore, a recent study showed that LCL161 preferentially radiosensitizes human papillomavirus-negative (HPV(−)) HNSC, providing justification for clinical testing of LCL161 in combination with radiation for patients with HPV(−) HNSC [140]. These studies demonstrated SMs as a promising strategy to counteract radiation resistance of cancer cells.

Nevertheless, there are a number of future challenges for the successful development of SMs as cancer therapeutics, as well as some drawbacks which emerged with the development of these compounds resulting from the complex cellular functions. Here, we used Strengths, Weaknesses, Opportunities, and Threats (SWOT) analysis to evaluate SMs (Figure 4). SWOT analysis is a business strategic tool to assess both internal (strengths and weaknesses) and external (opportunities and threats) factors. It is used extensively in business settings to uncover new outlooks and identify problems that would impede progress. The synthesis of the clinically useful SMs can be achieved depending on reasonable strategic planning.

Collectively, the above preclinical trials provide strong evidence that SMs can enhance the anticancer activity of TNFα, TRAIL, and chemotherapeutic agents, as well as radiosensitization in diverse tumor types; these findings suggest the therapeutic potential of such combinations for the treatment of human cancer. Accordingly, most subsequent clinical trials embarked on the use of SMs in rationally designed combination regimens.

### 4.3. Clinical Trial Development of SMs in Solid Tumors

So far, many SMs were identified, and eight SMs were evaluated in early clinical trials as cancer therapeutics, as listed in Table 1, including five monovalent compounds (LCL161, Debio 1143/AT-406/SM-406, CUDC-427/GDC-0917, RG7419/GDC-0152, and ASTX660) and three bivalent agents (birinapant/TL32711, APG-1387/SM-1387, and AEG40826/HGS1029). Chemical structures of the SMs are summarized in Figure 5. They were investigated as either a single agent or in combination setting with an anticancer agent in both solid tumor and hematological malignancies in terms of safety characteristics, maximum tolerated dose, pharmacokinetics (PK), pharmacodynamics (PD), biomarker identification, and initial efficacy.

Among the clinical trials listed in Table 1, limited published data of the RG7419/GDC-0152 clinical efficacy exist; however, it was the first SM to enter a phase I trial in June 2007, and it had the best profile among a series of compounds in the study [141]. Although no toxicity or efficacy was reported in the phase I clinical trial for RG7419 (NCT00977067), when administered intravenously to patients with locally advanced or metastatic malignancies, GDC-0152 demonstrated linear PK over doses ranging from 0.049 to 1.48 mg/kg.

The first human phase I trial of CUDC-427/GDC-0917 (NCT01226277), a second-generation, orally bioavailable SM from Genentech, showed that CUDC-427/GDC-0917 can be administered safely at doses up to 600 mg daily for 14 days every three weeks. The absence of severe toxicity, the inhibition of cIAP1 in peripheral blood mononuclear cell, and the antitumor activity warrant further studies [142]. Before that, modeling and simulation techniques were applied to study the preclinical suitability of the drug, and the simulations of human GDC-0917 plasma concentration–time profile and cIAP1 degradation at the 5-mg starting dose in the phase I clinical trial agree well with the observations. This work shows the value of preclinical studies in the early stages of the drug development process [143].

To date, seven studies which investigated the use of LCL161 as a solid tumor therapeutic were evaluated or are currently in the phase of early clinical trials. LCL161 was well tolerated on a weekly dosing schedule in 27 patients with advanced cancer (NCT01098838), and no dose-limiting toxicity was found at doses up to 1800 mg, administered as a single agent once weekly, in tablet formulation. This combined dose and formulation was well tolerated and had significant activity [144]. A total of 209 patients were enrolled in a randomized, phase II study of weekly paclitaxel with or without LCL161 (NCT01617668). The results indicated that the LCL161 with paclitaxel group showed higher pathological complete response rate (pCR) than the paclitaxel alone arm. Patients were randomized to the LCL161 1800 mg once weekly for 12 weeks; then, the LCL161 PK samples were taken on cycle 1 and cycle 4, where the median T_max_ was about 3.72 h and 3.50 h, respectively, while the median area under the curve from the time of dosing to the last measurable concentration (AUC_last_) was about 5250.70 ng∙h/mL and 5522.58 ng∙h/mL, respectively. In the overall study, more adverse events occurred in the combination arm compared to the paclitaxel alone arm, and pyrexia was the most common adverse event (19%) in the combination arm. In addition, two studies of LCL161 in combination with paclitaxel were completed, but no objective results were reported (NCT01968915 and NCT01240655). For the other studies, one is at the recruiting stage (NCT02890069), one is still active (NCT02649673), and one (NCT01934634) has an unknown status.

Debio 1143/AT-406/SM-406 is an oral agent currently being evaluated in six phase I or phase I/II trials on solid tumors [145]. The first trial (NCT01078649), a single-agent trial, was completed in 2014. Most common adverse drug reactions were fatigue (26%), nausea (23%), and vomiting (13%). One patient had a reversible elevation in aspartate transaminase (AST) which was the only dose-limiting toxicity (DLT) observed, and the maximum tolerated dose (MTD) was not reached. PK was dose-proportional above 80 mg without evidence of drug accumulation. Debio 1143 was well tolerated at doses up to 900 mg and elicited PD effects at doses greater than 80 mg. Based on these preliminary results, it has the potential to be used as an adjunctive treatment [146]. Then, in 2013, two trials of Debio 1143 in combination with common therapies were initiated. One was a phase I trial of Debio 1143 in combination with both paclitaxel and carboplatin, which is terminated according to their last update, but there are no results posted currently (NCT01930292). In another trial (NCT02022098), Debio 1143 is currently being tested in a phase I/II randomized study to determine the MTD, safety, PK, and antitumor activity in combination with concurrent chemoradiation therapy. Subsequently, there was a phase I trial (NCT03270176) of a dose-finding study of Debio 1143 in combination with avelumab after platinum-based therapy, and the estimated completion of the trial is May 2022. Recently, two new clinical trials were initiated, and they are currently recruiting; one is a phase I/II trial (NCT04122625) to assess the safety and efficacy of Debio 1143 in combination with nivolumab, and the estimated completion is 20 January 2023. The other phase I trial (NCT03871959) of pembrolizumab in combination with Debio 1143 was mainly initiated to determine MTD, recommended phase II dose (RP2D), and recommended dosing regimen and to obtain target engagement data, and it is expected to be completed in May 2021.

Notably, the new monovalent non-peptidomimetic ASTX660 from Astex Pharmaceuticals [147], which showed the ability to induce apoptosis and the inhibition of tumor growth in a preclinical trial [64], is currently being tested in a phase I/II clinical trial (NCT02503423). It is a dose-escalation phase I/II study of advanced solid tumors and lymphomas. Although the phase II part of the study is ongoing, the ASTX660 demonstrated a manageable safety profile, and it showed the evidence of pharmacodynamic and preliminary clinical activity at the 180-mg/day RP2D [148].

Birinapant/TL32711 is a second-generation bivalent SM manufactured by TetraLogic Pharmaceuticals, and its tolerability was improved [149]. Birinapant/TL32711 was advanced into several phase I/II trial as a single agent (NCT00993239, NCT01486784, and NCT01681368). The first human phase I trial with birinapant as a single agent on dose escalation safety (NCT00993239) showed that birinapant was well tolerated with the MTD of 47 mg/m^2^ with favorable PK/PD; however, at 63 mg/m^2^, the dose-limiting toxicities included headache, nausea, and vomiting, and two cases of Bell’s palsy (grade 2) were observed [150]. These results support the ongoing clinical trials of birinapant in patients with cancer. In a phase II trial, the effectiveness of birinapant was evaluated (NCT01681368). Eleven patients received birinapant treatment alone, after which accrual was terminated for lack of clinical benefit as a single agent in this small population. Noonan and his teammate emphasized the importance of studying single-agent PD and activity of birinapant to design the rational combination therapy for future clinical trials [151]. To date, seven trials of the combination with birinapant/TL32711 in studies of solid tumor were advanced. Specifically, combinations with other standard chemotherapeutics such as irinotecan or docetaxel (NCT01188499) and conatumumab (NCT01940172) were completed, while combinations with gemcitabine (NCT01573780), pembrolizumab (NCT02587962) as well as intensity-modulated radiation therapy (NCT03803774) are in progress to identify optimal synergistic combinations. Among them, only one phase I/II trial (NCT01188499) posted results, which exhibited that birinapant does not show good results in safety and tolerability. In addition, two trials were withdrawn for unknown reasons (NCT01766622 and NCT02756130).

AEG40826/HGS1029, with an undisclosed structure (proposed to be a bivalent SM), was well tolerated in patients with advanced solid malignancies and intravenous schedules with an MTD of 3.2 mg/m^2^ in a phase I trial (NCT00708006). HGS1029 induced rapid and sustained reduction of cIAP1 levels after a single dose of administration and showed evidence of apoptosis induction in patients [152]. Confirmed tumor regression was reported in a patient with colon cancer and two patients (NSCLC, adrenocortical cancer) had stable disease for more than six months. However, no further trials were reported.

Another bivalent SM APG-1387/SM-1387 developed by Ascentage Pharma company, which demonstrated a potent antitumor effect on nasopharyngeal carcinoma [153], was advanced into a phase I/II trial in patients with advanced solid tumors or hematologic malignancies to test the safety, tolerability, and PK and PD profile. The compound was used as a single agent or in combination with systemic anticancer agents (NCT03386526). To date, limited published data exist regarding the clinical efficacy of APG-1387/SM-1387 in solid tumors specifically.

Monovalent and bivalent SMs differ in their pharmacologic properties, as intravenous administration is necessary for bivalent compounds, whereas monovalent compounds are orally bioavailable. Many studies reported that bivalent SMs have a higher affinity than monovalent SMs, consequently providing them with a stronger antitumor activity [152,154]. However, it is not yet possible to compare the toxicity between monovalent and bivalent SMs, as well as their antitumor capacity in vivo, based on current clinical studies. So far, although various structurally optimized SMs were developed, researchers still need to investigate newer compounds in clinical trials.

## 5. Predicted Potential Markers of SM Precision Therapy

SMs entered clinical trials as a very promising anticancer therapy. However, cancer cell lines respond differentially to SMs, which makes the identification of predictive molecular markers for response to SMs necessary. As mentioned above, mechanistic studies revealed that expression of various factors such as caspase-8, TNFα, RIP1, c-FLIP, FADD, cIAP1/2, and XIAP is essential for SM-mediated tumor cell death. As studies demonstrated that SMs can also conduct TNFα-induced necroptosis, i.e., a caspase-independent cell death pathway, caspase activation might not be suitable for indicating SM activity. Similarly, TNFα, RIP1, c-FLIP, and FADD may serve important roles in contributing to SM resistance in tumor cell lines. Nevertheless, their expression levels cannot indicate SMs sensitivity, as SMs can induce tumor cells death through both TNFα-dependent and TNFα-independent pathways. Additionally, among these molecules, the level of anti-apoptotic proteins (e.g., cIAP1, cIAP2, and XIAP) is also not suited for predicting whether the treatment of SMs is effective or not. There are several reasons for this. Firstly, decreased expression levels of cIAP1 can be expected in non-tumor tissues, as well as sensitive and resistant malignancies. Secondly, as detailed above, SM treatment will be non-effective under conditions in which cIAP1 is absent and cIAP2 is expressed to high levels. Thirdly, cIAP2 initially degrades after treatment but rebounds and is refractory to subsequent degradation in cancer cells that cannot respond to SMs. Lastly, biomarkers predictive of response were analyzed in ovarian cancer, and it was found that cIAP1, XIAP, and caspase-9 are not positive markers indicating SM susceptibility [129].

cIAP2 itself is not a fitting biomarker of the SM treatment response. However, it is worth noting that cIAP2 knockdown, but not cIAP1 or XIAP knockdown, sensitized cancer cells to apoptotic cell death induced by SMs [83]. Moreover, cIAP1 expression levels are relatively high but are decreased more efficiently than cIAP2 with the treatment of SMs [66]. As cIAP2 degradation requires the presence of cIAP1, cIAP2 may become more stable when cIAP1 is depleted by SMs. Alternatively, elevated cIAP2 expression in certain cell lines renders resistance to SMs, suggesting that inactivation of cIAP2 may be a decisive factor in SM-induced cell death. Taken together, we speculate that regulators of cIAP2 (e.g., USP11) may serve as indicators of the expected efficacy of SMs. USP11 is a cIAP2-specific deubiquitylase, which selectively and specifically regulates cIAP2 stability independent of cIAP1 and XIAP. We utilized the UALCAN database to explore differences in the mRNA expression of USP11, between tumor and normal tissues in multiple cancers. As shown in Figure 1k, a total of 23 datasets were involved in the analysis. In total, 14 of 23 analyses revealed significant differences between cancer and normal groups. Among them, overexpression of USP11 was found in CHOL, colon adenocarcinoma (COAD), HNSC, liver hepatocellular carcinoma (LIHC), and pheochromocytoma and paraganglioma (PCPG), while under-expression was observed in bladder urothelial carcinoma (BLCA), breast invasive carcinoma (BRCA), glioblastoma multiforme (GBM), kidney renal clear cell carcinoma (KIRC), kidney renal papillary cell carcinoma (KIRP), Lung adenocarcinoma (LUAD), prostate adenocarcinoma (PRAD), thyroid carcinoma (THCA), and uterine corpus endometrial carcinoma (UCEC). The protein expression levels of USP11 in cancers are yet to be analyzed through more experiments. However, the question as to whether or not levels of cIAP2-specific deubiquitylase may represent as suitable marker for patient selection also remains to be determined in future studies. In addition, the feedback upregulation of cIAP2 could be due to the loss of cIAP1-mediated ubiquitination, activation of NF-κB signaling by SMs, or alterations in other signaling pathways, such as the PI3K/protein kinase B-serine/threonine kinase (Akt) pathway, which regulate cIAP2 expression, whereby molecules in these pathways may also be helpful to judge the expected efficacy of SMs.

## 6. Conclusions

The main goal of the research and development of drugs is to provide life-saving clinical medication. This review highlighted several mechanisms of decreased expression of SMAC/DIABLO in tumors and situations where there will or will not be therapeutic responses to SMs. Meanwhile, we also showed clinical studies of SMs, which manifested an acceptable safety of their use as a single agent or together with conventional or nonconventional drugs. As SMs are either directly involved in cell death pathways or strengthened the sensitivity of cancer cells for additional cytotoxic stimuli, the ultimate outcome of the treatment is the result of a complicated network of different effects. Thus far, SMs are very promisingly molecular targeted anticancer drugs, which have wide-ranging applications. However, there is still a long way to maximize the effectiveness of SMs in clinical therapy, such as minimizing its potential side effects, determining fitting predictive biomarkers, and drug combinations in individual cancers. A related consideration in the clinical development of SMs surrounds the consequences of its induction of NF-κB activation, whose possible adverse effects include increased levels of cytokines in normal tissues [155]. Researches showed that cytokine release syndrome was found to be dose-related and occurred at higher doses of SMs. Thus, work needs to be done to establish the optimal dosage. For the better development of SMs as therapeutic agents, experimental investigations are also needed to validate suitable indications of SM effectiveness as monotherapies. Thus far, patients that will likely respond to SM treatment are yet to be identified. In summary, there is no doubt of the enormous therapeutic potential of SMs in combination with death-inducing ligands, kinase inhibitors, chemotherapy, and radiotherapy. The ongoing development of novel SMs with stronger potency and selectivity is tempting to maximize their antitumor activity.

## Figures and Tables

**Figure 1 cells-09-01012-f001:**
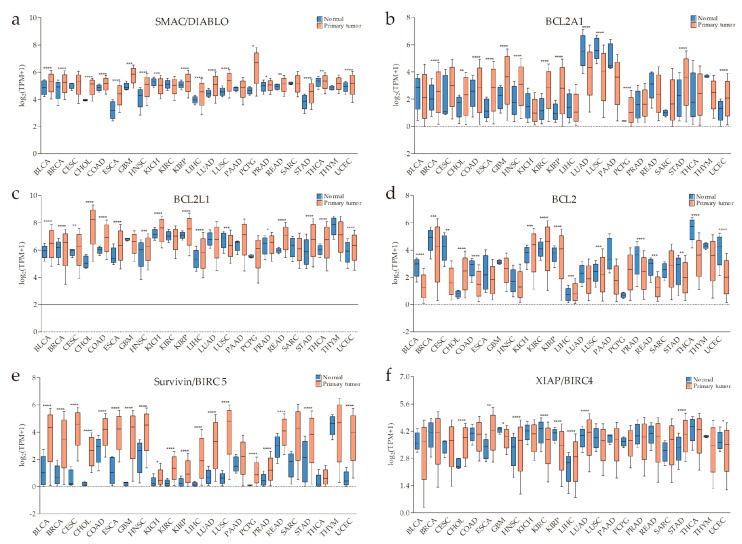
An overview of the expressions of second mitochondria-derived activator of caspase (SMAC)/direct inhibitor of apoptosis protein (IAP)-binding protein with low pI (DIABLO), apoptosis regulator BCL-2 (BCL-2) family members, baculoviral IAP repeat-containing protein 5 (survivin/BIRC5), IAPs, TNF-related apoptosis-inducing ligand (TRAIL), tumor necrosis factor (TNF), and ubiquitin-specific protease 11 (USP11) across The Cancer Genome Atlas (TCGA) cancers (with primary tumor and normal samples) from UALCAN (a comprehensive, user-friendly, and interactive web resource for analyzing cancer omics data). *X*-axis labeling is the category of cancer types. *Y*-axis is the logarithm of transcripts per million (TPM) of different samples. “TPM + 1” is used to avoid a situation where there is no solution when TPM is equal to zero. (**a**) SMAC is generally upregulated in the cancers listed. (**b**–**d**) The anti-apoptotic/pro-survival BCL-2 proteins, BCL2A1, BCL2L1, and BCL2 are significantly overexpressed in some types of tumors compared with normal tissues. (**e**) A pervasive high level of survivin in primary tumors. (**f**–**h**) High messenger RNA (mRNA) levels of IAPs in some types of cancer. (**i**–**k**) Increased TRAIL, TNFα, and decreased USP11 might be conducive to the effect of SMAC mimetics (SMs) on cancers (* *p* < 0.05, ** *p* < 0.01, *** *p* < 0.001, **** *p* < 0.0001).

**Figure 2 cells-09-01012-f002:**
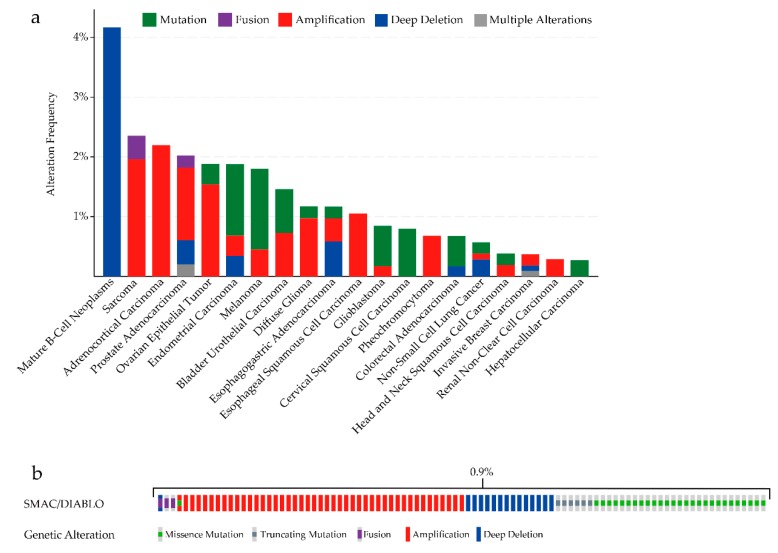
(**a**) SMAC/DIABLO mutation analysis in various types of human cancers (cBioPortal). (**b**) Oncoprint in cBioPortal representing the proportion and distribution of samples with alterations in SMAC/DIABLO. Cancers without mutations and samples without alterations are not shown in the figure.

**Figure 3 cells-09-01012-f003:**
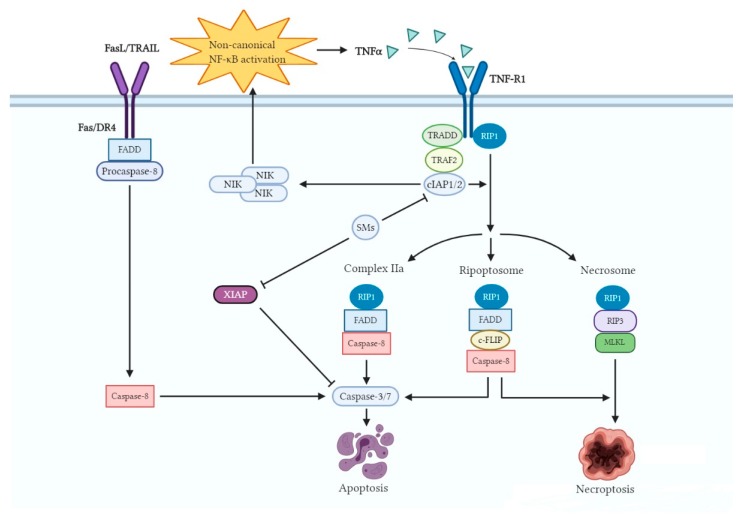
Regulation of cell death pathways by SMs. The binding of SMs to cIAPs, which enhances the E3 ubiquitin ligase activity of cIAPs, resulting in autoubiquitination and proteasomal degradation. These alterations lead to nuclear factor-κB (NF-κB)-inducing kinase (NIK) accumulation, non-canonical NF-κB activation, and tumor necrosis factor alpha (TNFα) production. The degradation of cIAPs raises three possible formations, i.e., complex IIa, ripoptosome, and necrosome, which induce cancer cell death. Meanwhile, SMs can directly bind XIAP, thereby activating caspases and inducing apoptosis.

**Figure 4 cells-09-01012-f004:**
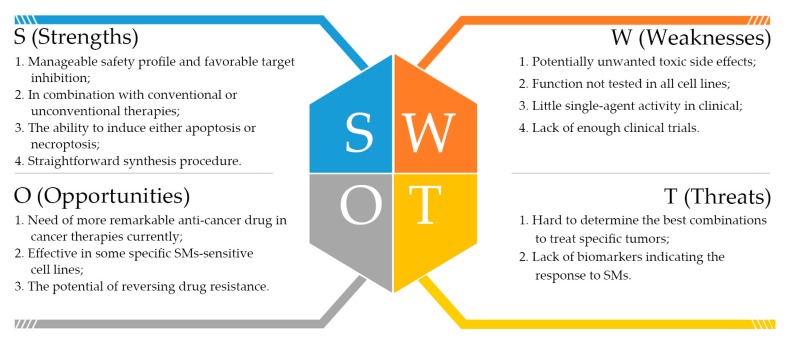
The Strengths, Weaknesses, Opportunities, and Threats (SWOT) analysis of SMs from our point of view.

**Figure 5 cells-09-01012-f005:**
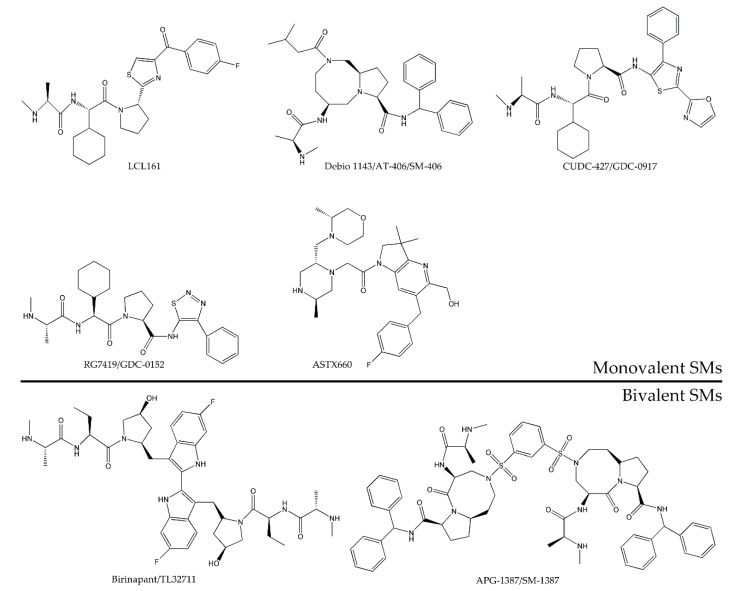
Chemical structure of monovalent and bivalent SMs tested in completed or ongoing clinical trials as anticancer agents.

**Table 1 cells-09-01012-t001:** SMs in completed or ongoing clinical trials.

Structural Class	SM Compound (Administration Route)	NCT Number	Phase	Status (As of 13 January 2020)	Cancer Type(s)	Drug(s) Combined	Trial Starting Date
**Monovalent**	**LCL161 (oral)**	NCT01098838	I	Completed	Advanced solid tumors	/	November 2008
		NCT01240655	I	Completed	Solid tumors	Paclitaxel	April 2011
		NCT01617668	II	Completed	Breast cancer	Paclitaxel	August 2012
		NCT01968915	I	Completed	Advanced solid tumors	Paclitaxel	November 2013
		NCT01934634	I	Unknown status	Metastatic pancreatic cancer	Gemcitabine and Nab-Paclitaxel	March 2014
		NCT02649673	I/II	Active, not recruiting	Small-cell lung cancer	Topotecan and pegylated granulocyte colony stimulating factor(PEG-GCSF)	23 March 2016
					Ovarian cancer		
		NCT02890069	I	Recruiting	Colorectal cancer	PDR001	14 October 2016
					Non-small-cell lung carcinoma (adenocarcinoma)		
					Triple-negative breast cancer		
					Renal cell carcinoma		
	**Debio 1143/AT-406/SM-406 (oral)**	NCT01078649	I	Completed	Cancer	/	29 March 2010
					Solid tumors		
					Lymphoma		
					Malignancy		
		NCT01930292	I	Terminated	Lung cancer	Paclitaxel and carboplatin	April 2013
					Ovarian cancer		
					Breast cancer		
		NCT02022098	Not Applicable	Active, not recruiting	Squamous cell carcinoma of the head and neck	Cisplatin and radiotherapy	October 2013
		NCT03270176	I	Recruiting	Non-small-cell lung carcinoma	Avelumab	10 October 2017
					Neoplasms		
		NCT04122625	I/II	Recruiting	Solid tumors	Nivolumab	8 April 2019
		NCT03871959	I	Recruiting	Adenocarcinoma of the pancreas	Pembrolizumab	13 September 2019
					Adenocarcinoma of the colon		
					Adenocarcinoma of the rectum		
	**CUDC-427/GDC-0917 (oral)**	NCT01226277	II	Completed	Solid tumors	/	October 2010
					Lymphoma		
	**RG7419/GDC-0152 (oral)**	NCT00977067	I	Terminated	Solid tumors	/	June 2007
	**ASTX660 (oral)**	NCT02503423	I/II	Recruiting	Solid tumors	/	July 2015
					Lymphoma		
**Bivalent**	**Birinapant/TL32711 (intravenous)**	NCT00993239	I	Completed	Refractory solid tumors	/	November 2009
					Lymphoma		
		NCT01188499	I/II	Completed	Advanced or metastatic solid tumors	Chemotherapy drugs	October 2010
		NCT01573780	I	Terminated	Unspecified adult solid tumor	Gemcitabine hydrochloride	April 2012
		NCT01681368	II	Terminated	Epithelial ovarian cancer	/	15 August 2012
					Peritoneal neoplasms		
					Fallopian tube neoplasms		
		NCT01766622	II	Withdrawn	Ovarian neoplasms	[^18^F]-CP18	30 November 2012
					Ovarian cancer		
					Fallopian tube neoplasms		
					Fallopian tube cancer		
		NCT01940172	I	Completed	Relapsed epithelial ovarian cancer	Conatumumab	November 2013
					Relapsed primary peritoneal cancer		
					Relapsed fallopian tube cancer		
		NCT02587962	I/II	Recruiting	Solid tumors	Pembrolizumab	4 August 2017
		NCT02756130	I/II	Withdrawn	High-grade fallopian tube serous adenocarcinoma	Carboplatin	1 August 2018
					High-grade ovarian serous adenocarcinoma		
					Primary peritoneal high-grade serous adenocarcinoma		
					Recurrent fallopian tube carcinoma		
					Recurrent ovarian carcinoma		
					Recurrent primary peritoneal carcinoma		
		NCT03803774	I	Recruiting	Recurrent head and neck squamous cell carcinoma	Radiation: intensity-modulated radiation therapy	7 January 2019
	**APG-1387/SM-1387 (intravenous)**	NCT03386526	I/II	Recruiting	Advanced solid tumors	/	21 November 2017
					Hematologic malignancies		
	**AEG40826/HGS1029 (intravenous)**	NCT00708006	I	Completed	Solid tumors	/	May 2008

Data were gathered by searching the National Institutes of Health (NIH)’s Clinical Trials.gov database at https://clinicaltrials.gov/. This table includes information on clinical trials as of 13 January 2020.

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
