# Peer review of "Potency and Selectivity of SMAC/DIABLO Mimetics in Solid Tumor Therapy"

_cells, 2020, doi:10.3390/cells9041012_

Round 1

Reviewer 1 Report

 This is a well organized review about SMAC/DIABLO Mimetics from bench to bedside. The authors present convincing evidence as to how these drugs may be promising choices for cancer treatment. The SWOT analysis is an interesting means to summarize their current status. Some of the opinions are overstated and lead to misleading or ambitious conclusions, that would require more data or tempering of the language.

Major comments:

  1. Page 18/29 last 2 lines, it is suggested that "USP11 plays a vital role in regulation of cIAP2", but the data only show a weak positive correlation between BIRC3 and USP11 at the mRNA level. without further supporting data, this should be modified to state "may play a role..." 
  2. Page 19/29 last line, states that "patients with lower mRNA levels of USP11 were predicted to have better efficacy of SMs", but there is no clinical or experimental evidence showing this response to SM treatment. Lee et al. [142] said only "over expression of USP11 led to stabilization of cIAP2 and promoted SMs therapeutic resistance", but this is the opposite of predicting response.

Minor comments:

    1. Figure 1. should be labeled "a" to "f" for each panel, to correspond with the legend.
    2. Page 6/29 line 3-4 [3.1. Development of SMs],  you described "bivalent SMs exhibited higher binding affinities as well as their ability to bind to the BIR2 and BIR3 domains of XIAP", but new monovalent SM, ASTX660 binds to the isolated BIR3 domains of both XIAP and cIAP1 with nanomolar potencies. ASTX660 is currently being tested in a phase I-II clinical trial (NCT02503423) and should be cited (Mol Cancer Ther. 2018 Jul;17(7):1381-1391).
    3. Page 8 last 6 lines, the SWOT analysis is interesting, but may not be recognizable to readers. Consider describing the origin of this analysis.
    4. In caption of Figure 5, "NF-kB" has an unusual character.
    5. Page 17/29, last 5 lines describes [147, 148], Beug et al.' s conclusion was "autocrine TNF-a production through SP3 essential for SMs induced apopotosis". This sentence should distinguish autocrine TNF-a from serum TNF-a.
    6. Please review grammar and word choice with a native English speaker throughout the manuscript.

Reviewer 2 Report

This is a poorly written and constructed review of SMAC mimetics. It needs extensive revision to be acceptable. The English is very poor. It adds nothing to the current understanding of the field and is not even a particularly useful or competent summary of what is known.

Some specific points:

In section 2.1, the statement that SMAC is generally upregulated in tumours needs qualified. It clearly isn't at the protein level.

Why show the in silico data for Blc-2 family genes (Figure 1) rather than all the IAPs? Only Survivin is shown. TNF, TRAIL expression more relevant too.

Contrary the way the authors have phrased it, the mutation rate of SMAC (Figure 2) is actually remarkably low.

The authors should say why we are interested in the effects of SMs on TRAIL-induced apoptosis? 

There is nothing about the effects of SMs on the immune microenvironment.

There is nothing discussed about why SMs might synergize with chemotherapy and radiotherapy, although this is known.

The section on the mechanism-of-action comes in too late. It is also superficial. A discussion on FLIP and the roles its different splice forms play in regulating apoptosis and necroptosis is needed. Also, as far as I am aware FLIP is not a part of the necrosome as depicted in Figure 5.

The SWOT analysis is unconvincing.

The review of clinical trials is comprehensive but too long.

There is no mention of Astex Pharmaceuticals new SM, which is the first non-peptidomimetic. It is in phase 2: NCT02503423. This agent could be a game changer for Smac mimetics as anti-cancer drugs but is not discussed.

While USP11 is a regulator of cIAP2, I don't see the relevance of the Kaplan-Meier curves in Figure 7. And the correlations between cIAP2 expression and USP11 expression in Figure 6 are very weak.

Round 2

Reviewer 1 Report

The authors have responded to all of my comments. I have no further concerns. 

Reviewer 2 Report

This is much improved over the original version that I reviewed.

Major

I still do not get the USP11 angle. The correlations in figure 6 between USP11 and cIAP2 are at the mRNA level, and if I understand correctly, USP11 is meant to stabilize USP11 at the protein level. The K-M curves correlating USP11 expression with survival in several cancers may have absolutely nothing to do with IAPs and are not as far as I can see from trials of Smac mimetics in those cancers; therefore, what is the point of this section? I would drop it in favour of recent studies such as that by Crawford et al in Cell Death & Diff which use a systems biology approach to predict response to Smac mimetics.

Minor

  1. The authors should give an indication of the percentage of cases in each cancer listed in the 2nd paragraph of page 2 in which SMAC expression is suppressed relative to normal tissues.
  2. Are any of the mutations identified in figure 2 in the AVPI domain, i.e. how many are functionally relevant?
